# Intrusion Detection in Intelligent Connected Vehicles Based on Weighted Self-Information

**Tianqi Yu [1]** **, Jianling Hu [1,2]** and **Jianfeng Yang [1,***]

1   School of Electronic and Information Engineering, Soochow University, Suzhou 215006, China; tqyu@suda.edu.cn (T.Y.); jlhu@suda.edu.cn (J.H.)
2   School of Electronic and Information Engineering, Wuxi University, Wuxi 214105, China
*   Correspondence: jfyang@suda.edu.cn

**Abstract:** The Internet of Vehicles (IoV) empowers intelligent and tailored services for intelligent connected vehicles (ICVs). However, with the increasing number of onboard external communication interfaces, ICVs face the challenges of malicious network intrusions. The closure of traditional vehicles had led to in-vehicle communication protocols, including the most commonly applied controller area network (CAN), and a lack of security and privacy protection mechanisms. Therefore, to protect the connected vehicles and IoV systems from being attacked, an intrusion-detection method is proposed based on the features extracted from the arbitration identifier (ID) field of CAN messages. Specifically, a sliding window is used to continuously extract a frame of streaming CAN messages first. Afterward, the weighted self-information of the CAN message ID is defined, and both the weighted self-information and the normalized value of an ID are extracted as features. Based on the extracted features, a lightweight one-class classifier, the local outlier factor (LOF), is used to identify the outliers and detect malicious network intrusion attacks. Simulation experiments were conducted based on the public CAN dataset provided by the HCR Lab. The proposed method, using four different one-class classifiers, was analyzed, and it is also benchmarked with three information entropy-based intrusion-detection methods in the literature. The experimental results indicate that, compared to the benchmarks, the proposed method dramatically improves the detection accuracy for injection attacks, namely denial-of-service (DoS) and spoofing, especially when the number of injected messages is low.

**Keywords:** network intrusion detection; one-class classifier; controller area network (CAN); intelligent connected vehicle (ICV)

## 1. Introduction

The upcoming 5G-Advance and 6G mobile communication networks can boost the development of the Internet of Vehicles (IoV). In the IoV, by the exploitation of embedded communications, sensing, and information-processing modules, intelligent connected vehicles (ICVs) can intelligently be aware of transportation environments and effectively exchange information with pedestrians, peer vehicles, and roadside infrastructures. The integration of communications, sensing, and computation facilitates ICVs with intelligent services such as advanced driver-assistance systems (ADAS) and autonomous driving [1,2].

Meanwhile, with the increasing number of embedded electronic control units (ECUs) and external communication interfaces, the in-vehicle network (IVN) has been developed from a simple point-to-point control bus to a distributed and heterogeneous communication and control network to guarantee controls and communications among the onboard modules [3]. A diagram of an ICV with a heterogeneous in-vehicle network in the IoV is provided in Figure 1.

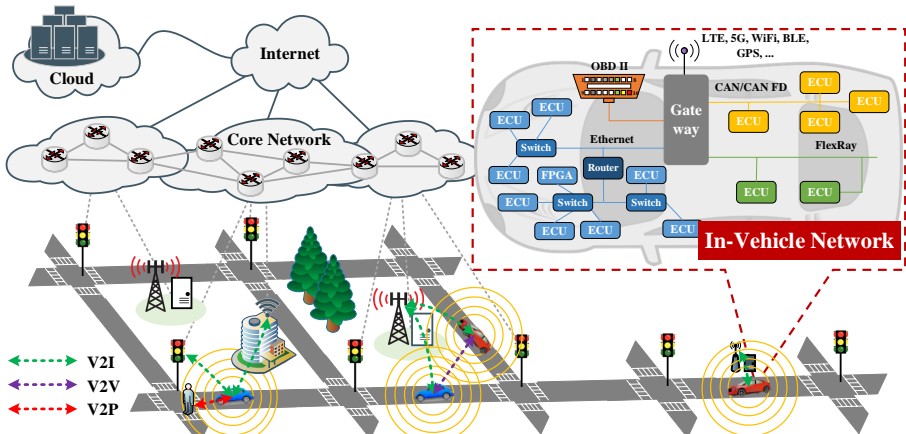

**Figure 1.** Diagram of an intelligent connected vehicle with a heterogeneous in-vehicle network in the Internet of Vehicles.

With external communication interfaces, the IoV can provide intelligent and tailored services to connected vehicles while bringing potential malicious network intrusion to the IVN. The closure of traditional IVN has resulted in the existing in-vehicle communication protocols, especially the most widely deployed controller area network (CAN), and a lack of security and privacy protection mechanisms, including access control, authentication, and encryption. Due to the lack of access control mechanisms, attackers can directly invade the IVN by cracking external communication interfaces. If malicious intrusion cannot be detected in time, the attacker can manipulate the compromised vehicle by controlling the CAN bus. For example, Miller and Valasek forced a Jeep Cherokee running on the highway to brake and rush to the roadside by remotely intruding on the CAN bus. Chrysler had to recall 1.4 million vehicles [4]. The Keen Security Lab of Tencent implemented the remote intrusion and absolute control of the CAN bus of Tesla S series vehicles in parking and driving states through a Wi-Fi interface, forcing Tesla to update its IVN system [5,6].

Moreover, if the compromised vehicle fails to detect malicious network intrusion in time, the intruder can invade and control other connected vehicles through the IoV. However, the cost of updating and manufacturing makes it difficult to replace current in-vehicle network architecture and communication protocols such as the CAN bus. At the same time, due to the limitations of in-vehicle network resources, the network security mechanisms used for computer networks are too complex to be applied to IVNs directly. Therefore, it is critical to develop intrusion-detection methods for IVNs based on existing network architecture and protocols to detect malicious network intrusions in a timely and accurate manner, thus preventing further attacks and threats.

Malicious intruders manipulate the compromised vehicles by injecting messages. Message injection can incur an abnormal pattern of CAN message information entropy, which is the information entropy of the CAN message arbitration identifier (ID) per unit of time. Therefore, some research efforts have been dedicated to ID entropy-based intrusion detection. Müter and Asaj detected flooding and injection attacks with a specific ID based on the fluctuations of the CAN message ID information entropy and relative entropy in a unit time window for the first time [7]. In [8], the authors further proposed the concept of relative distance to detect an injection attack with a legal ID. Using the information entropy of different message IDs in a unit time window as features, Wu et al. [9] proposed a novel sliding-window strategy with a fixed number of messages to avoid the interference of different baud rates and aperiodic CAN messages on the information entropy. Traditional message information entropy-based intrusion-detection methods can detect flooding attacks and other injection attacks with massive and high-frequency message injections. However, it can hardly detect attacks with few injected messages that have little impact on information entropy.

To resolve the above issues, a CAN message ID features-based intrusion-detection method is proposed in this work. First, a sliding window is used to continuously extract a frame of streaming CAN messages. Subsequently, the weighted self-information of the CAN message ID is defined, and both the weighted self-information and the normalized value of an ID are extracted as features. A lightweight one-class classifier, the local outlier factor (LOF), is then used to identify the outliers and detect malicious network intrusion attacks. Simulations have been conducted based on the public CAN dataset provided by the HCR Lab. The proposed method is analyzed using four different one-class classifiers, namely LOF, support vector data description (SVDD), isolation forest (iForest), and Ellipti-cEnvelope. The traditional information entropy-based intrusion-detection methods in the literature [7–9] are adopted as benchmarks. Experimental results indicate that, compared to the benchmarks, the proposed method dramatically improves the detection accuracy of injection attacks, namely denial-of-service (DoS) and spoofing, especially when the number of injected messages is low. The results also unveil that, considering the detection accuracy and the time complexity, LOF is the preferred one-class classifier for this work.

The rest of the paper is organized as follows. The structure of the CAN data frame is introduced in Section 2 first. Afterward, the CAN message ID features-based intrusion-detection method is described in Section 3. In Section 4, the performance of the proposed method is evaluated. Finally, the paper is concluded in Section 5.

## 2. CAN Data Frame

The structure of the CAN data frame is shown in Figure 2.

| Arbitration Field | | Control Field | | | Data Field | CRC Field | | ACK Field | | |
|---|---|---|---|---|---|---|---|---|---|---|
| **S O F** | **ID** | **R T R** | **I D E** | **R B O** | **D L C** | **Data** | **C R C** | **Del** | **A C K** | **Del** | **E O F** |
| 1bit | 11bits | 1bit | 1bit | 1bit | 4bits | 0~64bits | 15bits | 1bit | 1bit | 1bit | 7bits |

**Figure 2.** Structure of CAN data frame.

- SOF (Start Of Frame) and EOF (End Of Frame) indicate the range of the data frame.
- Arbitration ID is a unique identifier for destination receiver filtering and priority identifying. The lower ID indicates the higher priority.
- RTR (Remote Transfer Request) flags a remote frame.
- IDE (IDentifier Extension) flags the extension format.
- RB0 (Reserved Bit 0) is a reserved bit.
- DLC (Data Length Code) indicates the length of the data payload. Data field comprises the data payload of the message.
- CRC (Cyclic Redundancy Check) field checks the error of data transmission.
- ACK (ACKnowledgement) flags the normal CRC.

## 3. Intrusion Detection Based on CAN Message ID Feature Extraction

The concept of weighted self-information of a CAN message ID is clarified first, followed by the two-dimensional ID features-based intrusion-detection method.

### 3.1. Weighted Self-Information of CAN Message ID

The structure of the CAN data frame is shown in Figure 2. The message ID occupies 11 bits in the frame, which is used to identify the destination and priority. The lower value of ID indicates a higher priority. Within a time window, the probability of message ID being $i$ is calculated as

$$p_i = n_i / n_{all}, \tag{1}$$

where $n_i$ is the number of messages with ID $i$ within the time window, while $n_{all}$ is the total number of all the messages within the time window.

The self-information is thus determined by

$$I_i = -\log_2 p_i. \tag{2}$$

The weighted self-information of a message ID being $i$ is defined as

$$I_i^w = -p_i \log_2 p_i. \tag{3}$$

The entropy of message ID within a time window is calculated as

$$H(ID) = \sum_{i \in ID} I_i^w = \sum_{i \in ID} -p_i \log p_i, \tag{4}$$

where $ID$ is the set of all the message IDs showing up within the time window.

### 3.2. Local Outlier Factor (LOF)

The local outlier factor is a density-based unsupervised outlier detection method [10]. In this work, it is used as a one-class classifier, which can be applied to unknown attack detection. The LOF is detailed as follows.

#### 3.2.1. *k*-Distance

For a data point $\mathbf{x}_i$ in a $N-$sized dataset $\mathbf{X}$, the Euclidean distance to the rest of the data points in the same set is calculated by

$$dist(\mathbf{x}_i, \mathbf{x}_j) = \|\mathbf{x}_i - \mathbf{x}_j\|_2, \forall j \in [1, N], j \neq i, \tag{5}$$

where $\| \cdot \|_2$ is the $l^2$ norm.

The $k$-distance of data point $\mathbf{x}_i$ is denoted as $dist_k(\mathbf{x}_i)$. It is defined by there being at least $k$ data points in the rest of the set that meet the condition $dist(\mathbf{x}_i, \mathbf{x}_j) \leq dist_k(\mathbf{x}_i)$ and at most $k-1$ data points meet the condition $dist(\mathbf{x}_i, \mathbf{x}_j) < dist_k(\mathbf{x}_i)$.

Based on the definition of $k$-distance, the set of $k$-nearest neighbors (kNN) of the data point $\mathbf{x}_i$ is thus defined as

$$Nb_k(\mathbf{x}_i) = \{\mathbf{x}_j \in \mathbf{X}_{j \neq i} | dist(\mathbf{x}_i, \mathbf{x}_j) \leq dist_k(\mathbf{x}_i)\}. \tag{6}$$

Please note that $|Nb_k(\mathbf{x}_i)| \geq k$ as there is possibly more than one data point with the $k$th distance.

#### 3.2.2. Reachability Distance

The reachability distance (RD) from data point $\mathbf{x}_i$ to $\mathbf{x}_j$ is defined as

$$RD_k(\mathbf{x}_i, \mathbf{x}_j) = \max\{dist(\mathbf{x}_i, \mathbf{x}_j), dist_k(\mathbf{x}_j)\}, \tag{7}$$

where $dist(\mathbf{x}_i, \mathbf{x}_j)$ is the Euclidean distance between $\mathbf{x}_i$ and $\mathbf{x}_j$, and $dist_k(\mathbf{x}_j)$ is the $k$-distance of $\mathbf{x}_j$. Please note that the RD is directional, such that $RD_k(\mathbf{x}_i, \mathbf{x}_j)$ may not be equal to $RD_k(\mathbf{x}_j, \mathbf{x}_i)$.

#### 3.2.3. Local Reachability Density

Based on kNN and RD, the local reachability density (LRD) of data point $\mathbf{x}_i$ is given by

$$LRD_k(\mathbf{x}_i) = \frac{|Nb_k(\mathbf{x}_i)|}{\sum_{\mathbf{x}_j \in Nb_k(\mathbf{x}_i)} RD_k(\mathbf{x}_i, \mathbf{x}_j)}, \tag{8}$$

which evaluates the average reachability of $\mathbf{x}_i$ to its kNN.

### 3.2.4. LOF Score

The LOF score is finally defined as

$$LOF_k(\mathbf{x}_i) = \frac{1}{|Nb_k(\mathbf{x}_i)|} \sum_{\mathbf{x}_j \in Nb_k(\mathbf{x}_i)} \frac{LRD_k(\mathbf{x}_j)}{LRD_k(\mathbf{x}_i)}, \tag{9}$$

where the local reachability density of data point $\mathbf{x}_i$ is compared with that of its kNN. The LOF score is used for outlier detection,

$$Label(\mathbf{x}_i) = \begin{cases} inlier, & LOF_k(\mathbf{x}_i) \leq \delta, \\ outlier, & LOF_k(\mathbf{x}_i) > \delta, \end{cases} \tag{10}$$

where $\delta$ is the detection threshold determined by the specific applications.

### 3.3. Intrusion Detection Based on Extracted ID Features

A block diagram of the proposed ID features-based intrusion-detection method is depicted in Figure 3, and the pseudocode is listed in Algorithm 1. The specific procedures are provided as follows.

---

**Algorithm 1** ID Features-based Intrusion Detection

---

1: **Input:** streaming CAN message number $n = 0$, window size $n_{all}$, threshold $\delta$
2: **while** True **do**
3:   cumulate streaming CAN message $n = n + 1$
4:   **if** $n = n_{all}$ **then**
5:     # *ID* is the CAN message ID set of the time window
6:     **for** $i \in ID$ **do**
7:       calculate the weighted self-information of ID $i$ by (3) $\rightarrow I_i^w$
8:       normalize the ID $i$ by (11) $\rightarrow \bar{i}$
9:       calculate the LOF score of ID $i$ by (9) $\rightarrow LOF_k([I_i^w, \bar{i}])$
10:       **if** $LOF_k([I_i^w, \bar{i}]) > \delta$ **then**
11:         intrusion alert
12:       **end if**
13:     **end for**
14:     $n = 0$
15:   **end if**
16: **end while**

---

- A sliding window is used to accumulate several consecutive messages, where the window size, namely the total number of messages within a sliding window, is $n_{all}$.
- Features of the message IDs are extracted. In this work, two features are extracted. One is the proposed to be weighted self-information. The other is the normalized ID, which is calculated as

$$\bar{i} = i/0x07ff, \tag{11}$$

where $i \in ID$, and 0x07ff is the upper limit of CAN message IDs due to the pre-defined length of 11 bits.
- A lightweight one-class (OC) classifier LOF that takes the two-dimensional ID features as input is used to identify the abnormal messages incurred by the malicious network intrusion.

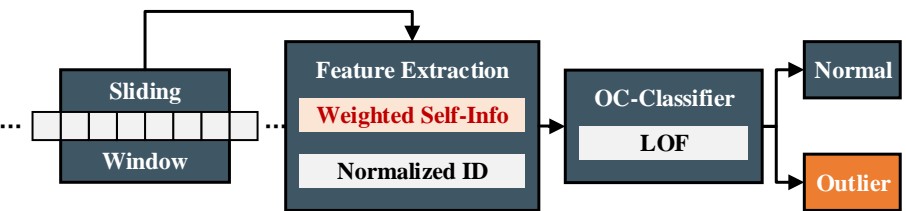

**Figure 3.** Diagram of the intrusion detection based on extracted ID features.

## 4. Performance Evaluation

In this section, the practical dataset used for simulation experiments and the metrics used for performance evaluation are introduced first. Subsequently, the simulation experiments are described, and an insightful analysis of the experimental results is provided.

### 4.1. Experimental Dataset

The public dataset provided by the HCR Lab of Korea University was adopted for the simulation experiments [11]. The dataset used for intrusion detection is described in Table 1. As illustrated in Table 1, we adopted 30,000 samples of the attack-free dataset that were collected from normal driving conditions. For the analysis of DoS and spoofing attack detection, 200 groups of attacks were randomly injected into the dataset, and the number of injected messages ranged from 5 to 50. A DoS attack floods the CAN using the message ID with the highest priority, i.e., 0x0000, to prevent normal communications and services. A spoofing attack pretends to be a normal ECU and sends messages with a legal ID, such as 0x0316, to manipulate the vehicles with malicious operations, such as urgent brake and acceleration. In the simulation experiments, attack-free data were used for classifier training and validation, where the ratios were 80% and 20%, and the data with attacks were used for testing. The window size was fixed at 50.

**Table 1.** Experimental Dataset.

|                 | Count          | ID Range           |
| --------------- | -------------- | ------------------ |
| Attack-free     | 30,000         | 0x0001~0x07ff      |
| DoS Attack      | 31,000~40,000  | 0x0000~0x07ff      |
| Spoofing Attack | 31,000~40,000  | 0x0001~0x07ff      |

### 4.2. Evaluation Metrics

The metrics used for evaluation are accuracy (*ACC*), precision (*PRE*), recall (*REC*), and an F1-score (*F1S*) [12,13].

$$ACC = (TP + TN)/(TP + FP + FN + TN), \tag{12}$$

$$PRE = TP/(TP + FP), \tag{13}$$

$$REC = TP/(TP + FN), \tag{14}$$

$$F1S = 2 * PRE * REC/(PRE + REC), \tag{15}$$

where *TP*, *FN*, *FP*, and *TN* refer to the true positive, false negative, false positive, and true negative results, respectively.

### 4.3. Experimental Results

To evaluate the performance of the proposed method, the traditional entropy-based method [7], sliding-window entropy-based method [9], and relative distance-based

method [8] were adopted as benchmarks. In terms of the lightweight one-class classifiers, except for the LOF used in this work, SVDD [14], iForest [15], and EllipticEnvelope [16] were also considered.

### 4.3.1. Analysis of Attack Group Size

The detection accuracy of a DoS attack and spoofing attack under different numbers of injected messages per attack group are provided in Tables 2 and 3, respectively. From Tables 2 and 3, we can come to the following conclusions.

- The relative distance-based method [8] can detect the spoofing attack with different attack group sizes accurately but can hardly be applied to the DoS attack detection due to its definition. In [8], the DoS attack is detected by the traditional entropy-based method [7].
- The detection accuracy of the DoS attack and spoofing attack using the benchmarks, namely traditional entropy-based method [7] and sliding-window entropy-based method [9], is below 90% when the attack group size is smaller than 20, jumping to around 95% when the attack group size increases to 30. The reason for this is that the traditional methods calculate the overall information entropy of all CAN messages within a time window. Thus, the methods can detect the intrusion with massive and high-frequency message injection but can hardly detect the intrusion with few injected messages that have little impact on the information entropy.
- The proposed method outperforms the benchmarks, especially when the number of injected messages is low. This is because the proposed method extracts the weighted self-information and normalized ID as features, which considers the information entropy of the messages with different IDs individually. Hence, it is more sensitive to the information entropy variation than the traditional methods considering the information entropy of all the messages.
- In terms of the proposed method with different one-class classifiers, the detection accuracy of DoS attack ranking in descending order is LOF > SVDD > EllipticEnvelope > iForest. The detection accuracy of spoofing attack ranking in descending order is LOF > SVDD > iForest > EllipticEnvelope.

**Table 2.** DoS attack detection accuracy vs attack group size.

| Attack Group Size | 5 | 10 | 20 | 30 | 40 | 50 |
|---|---|---|---|---|---|---|
| Traditional Entropy | 79.02 | 80.46 | 81.55 | 95.94 | 96.27 | 94.22 |
| Sliding-Window Entropy | 78.59 | 81.25 | 85.38 | 94.26 | 95.68 | 95.33 |
| Relative Distance | – | – | – | – | – | – |
| Proposed + LOF | 97.42 | 97.73 | 97.55 | 97.85 | 97.78 | 97.78 |
| Proposed + SVDD | 97.11 | 97.07 | 97.03 | 96.98 | 96.82 | 97.32 |
| Proposed + iForest | 83.46 | 82.52 | 84.95 | 82.32 | 81.49 | 80.72 |
| Proposed + EllipticEnvelope | 89.95 | 89.76 | 90.17 | 89.85 | 89.49 | 90.06 |

**Table 3.** Spoofing Attack Detection Accuracy vs Attack Group Size.

| Attack Group Size | 5 | 10 | 20 | 30 | 40 | 50 |
|---|---|---|---|---|---|---|
| Entropy | 78.03 | 78.93 | 80.10 | 97.15 | 96.46 | 95.15 |
| Sliding-Window Entropy | 73.46 | 79.31 | 82.86 | 94.36 | 95.84 | 94.18 |
| Relative Distance | 99.27 | 99.80 | 99.25 | 98.30 | 97.22 | 95.72 |
| Proposed + LOF | 98.33 | 98.52 | 98.33 | 98.62 | 98.53 | 98.59 |
| Proposed + SVDD | 96.28 | 96.87 | 96.53 | 96.82 | 96.52 | 96.20 |
| Proposed + iForest | 98.05 | 97.89 | 97.67 | 95.39 | 90.60 | 91.38 |
| Proposed + EllipticEnvelope | 89.65 | 89.78 | 90.00 | 89.73 | 89.38 | 89.09 |

4.3.2. Analysis of One-Class Classifier

To further analyze the performance of the proposed method with different one-class classifiers under the small-scale attack, the accuracy, precision, recall, and F1-score of DoS and spoofing attack detection under the attack group size of 5 are listed in Tables 4 and 5, respectively. For DoS attack detection, the average performance of these four metrics in descending order is LOF > SVDD > EllipticEnvelope > iForest. For spoofing attack detection, the average performance of these four metrics in descending order is LOF > iForest > SVDD > EllipticEnvelope.

The time complexity of one-class classifiers is compared in Table 6. It can be seen that the descending order of the complexity of these classifiers is SVDD > LOF > iForest > EllipticEnvelope.

Overall, in terms of detection accuracy, LOF and SVDD perform better than iForest and EllipticEnvelope. For time complexity, iForest and EllipticEnvelope are less complex than SVDD and LOF. Thus, there is a tradeoff between detection accuracy and time complexity. Considering the detection accuracy and the time complexity, LOF is the preferred one-class classifier for this work.

**Table 4.** DoS Attack Detection: Attack Group Size = 5.

| Classifier | Accuracy | Precision | Recall | F1-Score |
|---|---|---|---|---|
| LOF | 97.42 | 97.96 | 99.44 | 98.69 |
| SVDD | 97.11 | 99.93 | 97.05 | 98.49 |
| iForest | 83.46 | 99.63 | 83.1 | 90.54 |
| EllipticEnvelope | 89.95 | 99.16 | 90.26 | 94.56 |

**Table 5.** Spoofing Attack Detection: Attack Group Size = 5.

| Classifier | Accuracy | Precision | Recall | F1-Score |
|---|---|---|---|---|
| LOF | 98.33 | 98.87 | 99.45 | 99.16 |
| SVDD | 96.82 | 99.90 | 96.84 | 98.36 |
| iForest | 98.05 | 99.92 | 98.07 | 98.99 |
| EllipticEnvelope | 89.65 | 98.86 | 90.55 | 94.52 |

**Table 6.** Time Complexity of One-Class Classifier.

| Classifier | LOF | SVDD | iForest | EllipticEnvelope |
|---|---|---|---|---|
| Complexity | $\mathcal{O}(n^2)$ | $\mathcal{O}(n^3)$ | $\mathcal{O}(nlogn)$ | $\mathcal{O}(n)$ |

**5. Conclusions**

To protect connected vehicles and IoV systems from being attacked, a CAN message ID features-based intrusion-detection method was proposed in this work. First, a sliding window was used to continuously extract a frame of streaming CAN messages. Afterward, the weighted self-information of the CAN message ID was defined, and both the weighted self-information and the normalized value of an ID were extracted as features. Subsequently, a lightweight one-class classifier LOF was used to identify the outliers and detect the malicious network intrusion attack. Simulations were conducted based on a public CAN dataset. The proposed method was analyzed with four different one-class classifiers, namely LOF, SVDD, iForest, and EllipticEnvelope. The three traditional information entropy-based intrusion-detection methods were adopted as benchmarks. The experimental results indicated that the proposed method dramatically improved the detection accuracy of DoS and spoofing attacks compared to the benchmarks, especially when the number of injected messages was low. Furthermore, LOF was the preferred one-class classifier for the proposed ID features-based intrusion detection based on the analysis of the detection accuracy and time complexity.

**Author Contributions:** Conceptualization, T.Y. and J.H.; methodology, T.Y.; software, J.Y.; validation, T.Y., J.H. and J.Y.; formal analysis, T.Y.; investigation, J.H.; resources, J.H.; data curation, T.Y.; writing— original draft preparation, T.Y.; writing—review and editing, T.Y.; visualization, T.Y.; supervision, J.H.; project administration, J.H. and T.Y.; funding acquisition, T.Y. All authors have read and agreed to the published version of the manuscript.

**Funding:** This work was supported by the National Natural Science Foundation of China under Grant 62101373 and the Natural Science Foundation of Jiangsu Province under Grant BK20200858.

**Data Availability Statement:** Not applicable.

**Conflicts of Interest:** The authors declare no conflict of interest.

## Abbreviations

The following abbreviations are used in this manuscript:

| | |
|---|---|
| ACK | ACKnowledgement |
| ADAS | advanced driver-assistance systems |
| CAN | controller area network |
| CRC | cyclic redundancy check |
| DLC | data length code |
| ECU | electronic control unit |
| EOF | end of frame |
| ICV | intelligent connected vehicle |
| ID | identifier in arbitration field |
| IDE | identifier extension |
| iForest | isolation forest |
| IoV | Internet of Vehicles |
| IVN | in-vehicle network |
| LOF | local outlier factor |
| RB0 | reserved bit 0 |
| RTR | remote transfer request |
| SVDD | support vector data description |

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
