# Peer review of "Intrusion Detection in Intelligent Connected Vehicles Based on Weighted Self-Information"

_electronics, doi:10.3390/electronics12112510_

Round 1

Reviewer 1 Report

In this paper, a  CAN message ID features-based 6 intrusion detection method has been proposed. The topic is worthy of investigation, the introduction is well-written, and the readability of the paper is at a high level.

The method is based on a lightweight one-class classifier, i.e. local outlier factor (LOF), that uses two features - the proposed weighted self-information and the normalized ID. The algorithm is clearly presented, and the numerical results are presented for the public datasets. The achieved performance improvement (when compared to the concurrent methods) is significant, with reasonable complexity.

In section 3, an overview of the well-known machine learning methods is given. However, the presented identities and the corresponding descriptions are not novel, and the relevant references are not cited. Also, the operator ||.||_2  in Eq. (5) is not defined. LOF is defined in two lines - 116 and 137. Evaluation metrics used in section 4.2 are also well-known, and the corresponding reference should be given. This should be improved in the revised paper. 

It is not explained why two presented features are chosen. Also, it was written: "In the simulation experiments, the attack-free data are used for classifier training, and the data with attacks are used for testing". I understand that only the attack-free data are used for classifier training. However, in the first phase of the testing, a part of the attack-free data should be also used for classifier testing to avoid the overfitting of the model. Please, give a comment about this.

Reviewer 2 Report

The major issue with the paper is that it failed to contrast the proposed approach with respect to competing approaches. In the performance evaluation section, the simulation results are compared with several competing algorithms. The accuracy of detecting DoS attack is excellent compared with existing state of the art algorithms for small window sides. Unfortunately, no insight is provided regarding why the proposed algorithm can improve the accuracy by such a large margin (20%) for small window sides. 

I think a separate section should be provided to compare with the competing approaches and explain what has led to the superiority of the proposed methods for small window sizes. The use of a small window size obviously is more advantages because the attack can be detected sooner.

For research that targets a practical problem, the broader issue should be discussed. The CAN protocol was not designed with security in mind. It has no sender authentication, which obviously would lead to various attacks. In addition to outlier detection, remedial actions should be discussed. What the driver or the vehicle itself could do in cases of attacks? Disable CAN and switch to some predefined safe modes? Ultimately, the CAN protocol must be completely changed with sender authentication. 

While the sentences are mostly grammatically correct, they often read very awkwardly. Even the title is extremely awkward. I suggest changing the title to: "Intrusion Detection in Intelligent Connected Vehicles Based on Weighted Self-Information."

Section 3 title also reads very awkward. How about changing it to "Feature Extraction Based on CAN Message ID"?

In line 198, "...performs stabily over..." also reads very awkward. I guess you meant the performance is insensitive to the window sides. Again, you should explain why the algorithm is insensitive.

Reviewer 3 Report

­The manuscript is devoted to a new method for detection of intrusion. The proposed method is based on controller area network message ID features.

Preliminaries to the research area are provided. In particular basic information about the connected vehicles and the risks of potential malicious network intrusions to the in-vehicle networks are given.

The authors introduce the structure of the constructed controller area network data frame. The proposed intrusion detection method is described in detail. The performance of the method is evaluated by the use of four classifiers. The results are compared with other approaches.

The presentation of the main results is clear. From a formal point of view, all the contents seems to be correct. The results are valuable and worthy of being published taking into account their possible applications in areas related to security and protection of intelligent connected vehicles.  

Minor revisions are suggested to improve the quality of the exposition:

p. 1, line 6: I suggest the meaning of all abbreviation to be given, in particular for ID too.

p. 5, line 137: I suggest commas to be added at the end of the equations.

Round 2

Reviewer 2 Report

The authors have fully addressed my issues. I do not have further comments.